# Exploratory Study of the Effect of IMA950/Poly-ICLC Vaccination on Response to Bevacizumab in Relapsing High-Grade Glioma Patients

**DOI:** 10.3390/cancers11040464

**Published:** 2019-04-02

**Authors:** Emma Boydell, Eliana Marinari, Denis Migliorini, Pierre-Yves Dietrich, Anna Patrikidou, Valérie Dutoit

**Affiliations:** 1Laboratory of Tumour Immunology and Department of Oncology, Geneva University Hospital, 1211 Geneva, Switzerland; Emma.Boydell@etu.unige.ch (E.B.); eliana.marinari@unige.ch (E.M.); dmig@pennmedicine.upenn.edu (D.M.); 2Translational Research Center for Oncohaematology, Department of Internal Medicine Specialties, University of Geneva, 1211 Geneva, Switzerland; pierre-yves.dietrich@hcuge.ch; 3Department of Oncology, Clinical Research Unit, Dr Dubois Ferrière Dinu Lipatti Research Foundation, Geneva University Hospital, 1211 Geneva, Switzerland; anna.patrikidou@unige.ch

**Keywords:** glioma, bevacizumab, immunotherapy, IMA950, peptide vaccine

## Abstract

Immunotherapy, including therapeutic vaccines, is increasingly being developed for patients with high-grade glioma, and combinations of immunotherapies and synergy with standard of care are being investigated. In this regard, bevacizumab (BEV) has been shown to synergize with immunotherapy in preclinical studies of glioma and in other tumour entities. Here, we conducted a post-hoc exploratory study to evaluate the effect of the IMA950/poly-ICLC peptide vaccine on subsequent BEV administration in high-grade glioma patients. 16 IMA950-vaccinated and 40 non-vaccinated patients were included. At initial diagnosis, patients benefited from surgery and chemoradiation. At first or subsequent recurrence, patients received 10mg/kg of BEV every 2–3 weeks. Primary endpoints were overall survival (OS) and progression-free survival (PFS) from BEV initiation. IMA950-vaccinated patients did not show improved response to BEV as compared to non-vaccinated patients: there was no difference in median PFS (2.6 vs. 4.2 months for vaccinated and control patients, respectively, *p* = 0.50) nor in median OS (7.8 vs. 10.0 months for vaccinated and control patients, respectively, *p* = 0.69). In conclusion, potential synergy of BEV and therapeutic vaccines, when administered sequentially, has yet to be established in the clinical setting of GBM recurrence. Potential synergy of concomitant administration should be tested in future trials.

## 1. Introduction

Glioblastoma multiforme (GBM), or WHO grade IV astrocytoma, is an aggressive cancer of the central nervous system (CNS) arising from astrocytes. The current standard treatment for GBM is surgical resection followed by concurrent radiotherapy and chemotherapy with temozolomide (TMZ), followed by adjuvant TMZ monotherapy [1]. Despite therapeutic advances, prognosis remains poor with a median OS of 14.6 months for newly diagnosed GBM [1,2]. Recently, improvement has been reported with the use of tumour-treating fields in association with chemoradiation (median OS of 20.9 months), although the benefit remains controversial [3,4]. Isocitrate dehydrogenase (IDH) wildtype (wt) diffuse WHO grade III anaplastic glioma is another aggressive glioma subtype, with identical clinical management to that of GBM [5].

Efficacy of existing therapy for newly diagnosed high-grade glioma patients is limited, due to early tumour dissemination in the brain, tumour heterogeneity and rapid progression. Therefore, novel treatments are under investigation, among which therapeutic vaccination has shown promising results [6,7], despite the intrinsically immunologically challenging environment of CNS tumours [8]. The IMA950 multipeptide vaccine contains 11 glioma-associated antigens among which 9 are HLA-A*0201-restricted peptides, and 2 are HLA class II-binding peptides, ensuring stimulation of an integrated CD8^+^ and CD4^+^ T cell response [9]. Importantly, the IMA950 CD8 antigens were identified via peptide elution from the surface of GBM samples, ensuring antigen presentation to the immune system [9]. In addition, the IMA950 vaccine incorporates several antigens, limiting immunoediting and immune evasion [10,11]. Safety of IMA950 plus poly-ICLC, a synthetic TLR3 ligand, has been demonstrated in a phase I/II trial performed in Geneva and immunogenicity was demonstrated, with elicitation of CD8 and CD4 T cell responses in the majority of patients [6]. Safety and immunogenicity of IMA950 combined with GM-CSF has also been demonstrated in a phase I trial [7].

After initial treatment, tumour progression in high-grade glioma is inevitable. To date, no standard of care exists for recurring GBM and protocols are largely dictated by local practice. Treatment includes surgical re-intervention, radiotherapy, chemotherapy with cytotoxic agents (lomustine, irinotecan, etc.), and combinations of these therapies. In this regard, the high level of vascular proliferation and increased expression of vascular endothelial growth factor (VEGF) make high-grade glioma ideal targets for antiangiogenic drugs. BEV, a monoclonal anti-VEGF-A IgG_1_ antibody, was granted accelerated approval by the FDA in 2009, and subsequently authorized in Switzerland for relapsing GBM following two single-arm phase II trials in the recurrent GBM setting [12,13]. The European Medicines Agency declined approval due to the lack of a non-BEV control arm, modest improvement in OS, and challenges with radiographic response assessment [14]. Nonetheless, BEV seems to improve overall quality of life, perhaps associated with reduced glucocorticoid requirement [15].

The rationale supporting the combination of immunotherapy and antiangiogenic drugs is strongly established in preclinical settings [16]. The VEGF pathway induces immunosuppression through direct interaction with the immune system, by stimulating regulatory T lymphocytes, limiting antigen presentation by DC and inhibiting effector T cells [17,18,19,20,21]. Indirect interactions with the immune system also occur through modulation of tumour vasculature, VEGF blockade increasing T cell recruitment [19], and vessel normalisation improving access of therapeutic agents to the tumour site [22]. Conversely, it has been shown that immune stimulation at the tumour site results in vessel normalisation [23,24]. Clinical trials in non-GBM solid tumours have combined antiangiogenic drugs and immune checkpoint blockade (ICB). In metastatic melanoma, BEV addition to ipilimumab led to a substantial increase in patient survival [25]. Histological findings from this study demonstrated increased CD8 T cell and macrophage infiltration at the tumour site, and increased circulating memory T cells in peripheral blood in the ipilimumab/BEV combination arm.

Only recently have clinical trials started testing the synergy of antiangiogenic drugs and immunotherapy in recurrent GBM. An ongoing phase I trial testing hypofractionated stereotactic irradiation with BEV and pembrolizumab reported OS rates of 94% at 6 months and 64% at 12 months [26]. An ongoing phase II trial studying the effect of BEV addition to pembrolizumab reported no difference in survival between pembrolizumab/BEV association as compared to survival previously reported for BEV monotherapy [27]. In addition, trials testing therapeutic peptide or heat-shock protein vaccination with concomitant BEV administration are ongoing in patients with recurrent glioma [28,29,30]. The phase I/II clinical trial of IMA950 vaccine plus poly-ICLC in combination with standard of care in newly diagnosed GBM [6] offered a unique opportunity to study the potential synergy of BEV and a therapeutic vaccine, as BEV is largely used in Switzerland as a second- or further-line treatment for recurring GBM. We hypothesised that IMA950/poly-ICLC vaccination improves response to subsequent BEV administration after tumour relapse. To test this, we conducted an exploratory analysis comparing response to BEV in IMA950/poly-ICLC-vaccinated and in non-vaccinated patients.

## 2. Results

We hypothesized that IMA950/poly-ICLC vaccination improves clinical response to subsequent BEV administration after tumour relapse. To test this, we conducted an exploratory analysis comparing response to BEV in IMA950/poly-ICLC-vaccinated and in non-vaccinated patients. Patients from the cohort of the IMA950/poly-ICLC phase I/II trial were included [6]. We subsequently identified a control group from high-grade glioma patients in our centre treated with BEV after initial standard of care, without prior IMA950 therapeutic vaccination. Both groups were tested for homogeneity, and were compared using two primary endpoints, OS and PFS from BEV initiation.

### 2.1. Study Participants

After initial medical record reviewing, 56 patients were included in the analysis cohort: 40 patients in the control group and 16 patients in the IMA950 vaccination group. Thirty-eight patients were excluded. Main reasons for exclusion are listed in Figure 1.

### 2.2. Patient, Tumour and Treatment Characteristics

Patient characteristics are summarised in Table 1. The overall cohort included 38 men and 18 women, with an overall median age at BEV initiation of 59 years, without any significant difference in distribution between the groups. Regarding tumour parameters, five (12.5%) patients in the control group and two (12.5%) in the IMA950 group were initially diagnosed with grade III anaplastic astrocytoma, whilst the remaining patients, consisting the large majority of patients, were treated for primary GBM.

O6-Methylguanine-DNA-methyltransferase (MGMT) promoter methylation and IDH mutation status were systematically analysed in patients included in the IMA950 trial. MGMT methylation status differed between the two patient groups (*p* = 0.005 Chi-square test), with a significantly larger percentage of unmethylated patients in the IMA950 patient group (87.5% in IMA950 vaccinated vs. 37.5% in control patients). The large percentage of patients (42.5%) in the control group for whom MGMT promoter methylation status was unknown probably accounts for this difference. Further comparison taking only into account patients with known MGMT status showed similar rates of MGMT methylation between control and IMA950 patients (*p* = 0.27, Chi-square test). Only 10.0% of control patients and 6.0% of IMA950 patients displayed IDH mutated tumours, without a significant difference across the groups (*p* = 0.16, Chi-square test). Overall, every patient benefited from standard of care treatment. Adjuvant TMZ was administered to 36 (90.0%) patients in the control group and to all patients in the IMA950 group (*p* = 0.19). Adjuvant TMZ was not administered to patients showing unequivocal progression after chemoradiation. All patients received BEV as a subsequent treatment line upon progression (10 mg/kg intravenously every 2 to 3 weeks), without significant difference in exposure between groups (median 10.5 injections in both groups, *p* = 0.77, Mann-Whitney *U* test). Treatment modalities and combinations at tumour recurrence varied between patients, and included additional treatment lines before BEV, concomitant treatment with BEV (lomustine, irinotecan, TMZ), and further salvage therapies after progression under BEV (lomustine, irinotecan, re-irradiation, surgery, see Appendix A). Generally, in cases of BEV monotherapy, disease progression resulted in addition of another systemic line to the existing BEV treatment (mainly lomustine and irinotecan). The number of patients receiving identical treatment combinations being insufficient to allow further stratification, patients were grouped depending on whether BEV was introduced as a monotherapy or as a combination therapy upfront. 25 (62.5%) patients in the control group and 12 (75%) patients in the IMA950 group benefited from BEV monotherapy, and overall, BEV was administered as a second line in 42 (75.0%) patients of the total cohort. However, the time point of BEV administration (second or later-line treatment) did not significantly vary between the two groups. Further testing for heterogeneity between the two groups included classification in three prognostic classes according to the modified recursive partitioning analysis (RPA) by the Radiation Therapy Oncology Group (RTOG) [31]. There was no significant difference in the two patient groups with regard to RPA prognostic groups (Table 1).

### 2.3. Endpoints

All IMA950 patients progressed under BEV, and two patients were censored for OS. In the control group, four patients were censored for PFS and eight for OS. Median OS from BEV initiation was 7.8 (CI 6.9–8.7) months in the IMA950 group, versus 10.0 (CI 7.4–12.6) months in the control group, with no significant difference between the two (*p* = 0.69, log-rank test). Median PFS from BEV initiation was 2.6 (CI 0.4–2.9) months in the IMA950 group versus 4.2 (CI 2.8–5.5) months in the control group, with no significant difference between the two (*p* = 0.50, log-rank test, Table 2 and Figure 2). Survival rates at 6 months (PFS-6) was 30% for the control group and 25% for the IMA950 group. Survival rates at 9 months (PFS-9) and 12 months (PFS-12) were 22.5% vs. 18.8%, and 7.5% vs. 12.5%, respectively, with overall no difference between the groups (Table 2).

In order to detect potential difference in survival from diagnosis in the two groups, OS from histological tumour diagnosis was compared between the two groups. Median OS from diagnosis was 19.0 (CI 18.0–20.0) months in the IMA950 group versus 25.0 (CI 21.6–28.4) months, without significant differences (*p* = 0.53, log-rank test, Table 2). Altogether, we observed no significant differences in OS and PFS for patients receiving BEV with or without prior IMA950/poly-ICLC vaccination.

## 3. Discussion

This single-centre exploratory study focused on the potential interaction between therapeutic vaccination and BEV antiangiogenic treatment. We compared patient survival and disease progression between patients pre-treated with the IMA950/poly-ICLC peptide vaccine, and those who were not. Our results suggest that, in a homogenous cohort, prior IMA950 peptide vaccination does not alter subsequent response to BEV.

Although sample size was small, we successfully demonstrated lack of heterogeneity between the IMA950 vaccinated and the control group. No notable bias was identified in patient selection for the cohort. Patients included in the IMA950 group were HLA-A*0201^+^; however no evidence suggests that the HLA status affects glioma progression or response to BEV. Although MGMT status was unknown for a large number of patients in the control group, it should be noted that MGMT gene promoter methylation is a favourable prognostic factor that confers better response to TMZ chemotherapy, with no evidence for interference with BEV treatment [32]. Patients were also balanced with regard to IDH status: IDH-mutant GBM are associated with longer survival, and are often the result of progression from lower grade gliomas [4]. Patient classification according to the RTOG modified RPA showed no difference in clinical prognostic factors between groups. Consistently, OS in the two groups was superior to the reported OS per RPA class from the RTOG database reflecting treatment advances since the publication of the RPA classification, including extended use of re-irradiation, reoperation and BEV itself [31].

OS, which takes into account the effect of salvage therapies and PFS, which also accounts for stable disease, were selected as endpoints based on their clinical meaningfulness. Further endpoints such as objective response rate (ORR), which has the advantage of assessing tumour response to the investigated drug independently from previous and subsequent lines could also be analysed [33]. However, ORR does not account for stable disease, and therefore does not evaluate full clinical benefit in recurring GBM. The impact of BEV on quality of life could also have been measured, with indicators such as performance scores and neurological status. However, the post-hoc nature of our study did not allow for that.

Previously reported survival under BEV as a monotherapy or with concurrent chemotherapies (lomustine or irinotecan) is variable, with OS ranging from 4–12 months and PFS from 2.3–6 months [12,13,34,35,36,37]. The two single-arm phase II studies that enabled accelerated approval of BEV for recurrent GBM in 2009 reported survival under BEV monotherapy as 9.2 months OS and 4.2 months PFS [12], and 7.8 months OS and 4.0 months PFS, respectively [13]. In the combined BEV-irinotecan treatment, OS was 8.7 months and PFS was 5.6 months [12]. In the BELOB phase III trial, median OS and PFS were 8 and 3 months with BEV monotherapy, and 12 and 4 months for the BEV/lomustine combination [34]. More recently, a randomized phase III trial comparing survival in BEV-lomustine combination therapy versus lomustine monotherapy in recurrent GBM, reported median OS at 9.1 months, and PFS of 4.2 months in patients receiving combined lomustine and BEV [36]. Taken in consideration the results of these studies, pre-treatment with IMA950 vaccine, for which OS and PFS were 7.8 and 2.6 months respectively, does not seem to increase response to BEV after tumour relapse. Identifying whether IMA950-specific immune responses interact with VEGF pathways and whether measured levels of T cell responses are clinically significant will help to understand the absence of synergy between BEV and IMA950. The majority of patients receiving IMA950/poly-ICLC vaccination were able to elicit CD4 and CD8 T cell responses to the vaccine peptides [6]. However, neither presence nor magnitude of the T cell response to the vaccine did correlate with PFS or OS from BEV initiation. Analysing the pharmacokinetics of BEV, which does not cross the blood-brain barrier and therefore acts on endothelial VEGF receptors might as well be informative [38]. It is also important to note that IMA950/poly-ICLC vaccination and BEV were administered sequentially. As there is a strong rationale for combination of immunotherapies and BEV, future trials should test administration of therapeutic vaccines with concomitant BEV in patients with recurrent glioma. Finally, studies combining therapeutic vaccination with concomitant ICB such as pembrolizumab might show clinical synergy between immunotherapy and BEV by further increasing T cell response [39] and improving vessel normalisation [24].

In summary, this study did not show improved response to treatment by BEV in recurrent GBM patients previously vaccinated with IMA950. A major limitation of the current study is the small sample size in the IMA950 vaccination cohort. Further studies, with larger sample sizes, with concomitant administration and combinations with other immunotherapeutic agents might potentially show increased clinical benefit under antiangiogenic drugs and provide new prospective treatment modalities for recurrent GBM.

## 4. Materials and Methods

### 4.1. Study Design

This was an post hoc exploratory analysis using data from the previously reported IMA950 peptide-based vaccine with poly-ICLC adjuvant phase I/II trial (NCT01920191) [6], aiming to assess the efficacy and potential synergy of BEV on vaccine-pre-treated patients. To test this effect, survival data were compared to those of a balanced control group treated with BEV without vaccine treatment.

### 4.2. IMA950 Vaccine Treated Patients

The vaccinated patient cohort analysed in our study consisted of patients enrolled in the IMA950 peptide-based vaccine with poly-ICLC adjuvant phase I/II trial (NCT01920191) [6]. This clinical trial included 19 patients according to the following eligibility criteria: histological diagnosis of GBM or AA III, age > 18 years, HLA-A*0201^+^, ECOG performance status of 0 or 1 and a stable or decreasing steroid treatment regimen with a maximum of 4mg/day of dexamethasone. Main exclusion criteria were history of cardiac disease and HBV, HCV or HIV positive serology. IMA950/poly-ICLC was administered during an initial induction phase after chemoradiation (8 weeks after surgery), and subsequently during a maintenance phase concurrently with adjuvant TMZ (Figure 3). The IMA950 vaccine was composed of nine HLA-A*0201-restricted glioma-associated peptides, 2 HLA-DR-binding tumour-associated peptides and one HLA-A*0201-restricted HBV-derived peptide as marker of vaccination efficiency [6]. Six patients received the IMA950 vaccine (413 µg of each peptide, 4.96 mg total) intradermally and poly-ICLC (1.5 mg) intramuscularly. The remaining patients received IMA950/poly-ICLC mixed and injected subcutaneously (*n* = 7) or intramuscularly (*n* = 6) [6]. Patients received nine (range: 4–11) injections of the vaccine. The trial (NCT01920191) was conducted under the control and monitoring of the Swiss regulatory authorities (Swissmedic), as well as of the local Institutional Review Board and Ethics committee supervision. Signed informed consent was obtained for all patients. Sixteen of these patients received BEV (10 mg/kg every 2–3 weeks) at a time point later to standard first-line therapy and vaccination (Figure 3). Treatment modifications for TMZ and BEV were as per standard clinical protocols and national guidelines. No dose modifications were allowed for the IMA950/poly-LCLC vaccination.

### 4.3. Control Group

We retrospectively collected data from patients treated for primary GBM or secondary GBM, which include relapsing grade III glioma, at Geneva University Hospitals, Switzerland, between January 2009 and March 2018. Forty of the control patients benefited from BEV (10 mg/kg, every 2–3 weeks) without having received a prior vaccine treatment (Figure 3). Patients were excluded if progression occurred after initial diagnosis of a low-grade glioma without histological documentation of at least grade III glioma, and if first line treatment did not consist of surgery followed by chemoradiation with TMZ (i.e., the standard of care). Treatment modifications for BEV were as per standard clinical protocols and national guidelines.

### 4.4. Data Collection

Data collection was conducted in accordance with the Declaration of Helsinki and with approval of the local research ethics committee. No patient identifiable data were used. Patient characteristics, tumour characteristics, prior treatment modalities and subsequent salvage therapies after progression under BEV were extracted from medical records. Data collection included assessment of two molecular prognostic factors: IDH mutation and MGMT methylation status. The modified RTOG RPA [31] was used to assign analysed patients to a prognostic group at diagnosis depending on age, extent of surgical resection, baseline Karnofsky Performance Score (KPS) and neurological assessment (able to work vs. not able to work).

### 4.5. Endpoints

Primary endpoints were OS and PFS under BEV. OS was defined as time from BEV initiation to death; PFS was defined as time from BEV initiation to radiological or clinical evidence of tumour progression or death. Radiological progression was determined as per the RANO criteria with the adaptation for antiangiogenic therapy [40,41]. Secondary endpoints were PFS rates at 6, 9 and 12 months and OS from diagnosis, defined as time from histologic diagnosis to death. Surviving patients were censored at date of last follow-up for OS analysis. Surviving patients that did not show signs of progression were censored for PFS analysis at time of last disease assessment.

### 4.6. Statistical Analysis

The two patient cohorts were compared using Chi-square tests for categorical variables, and Mann-Witney *U* tests for continuous variables. Survival analysis was performed using the Kaplan-Meier method and compared by using log-rank analysis. Patients alive and not progressing at last follow-up were censored for the OS and PFS analysis, respectively. PFS-6, -9 and -12 were compared using Chi-square tests. *p*-values < 0.05 were considered statistically significant. Bonferroni correction was applied to *p*-values in subgroup comparisons. All tests were performed with IBM SPSS v.24 (IBC Inc., Armonk, NY, USA).

## 5. Conclusions

In summary, this study did not show improved response to subsequent treatment by BEV in recurrent GBM patients previously vaccinated with IMA950/poly-ICLC. Further studies, testing concomitant BEV/vaccine administration and combination with other immunotherapeutic agents will potentially demonstrate increased clinical benefit under antiangiogenic drugs and provide new prospective treatment modalities for recurrent GBM.

## Figures and Tables

**Figure 1 cancers-11-00464-f001:**
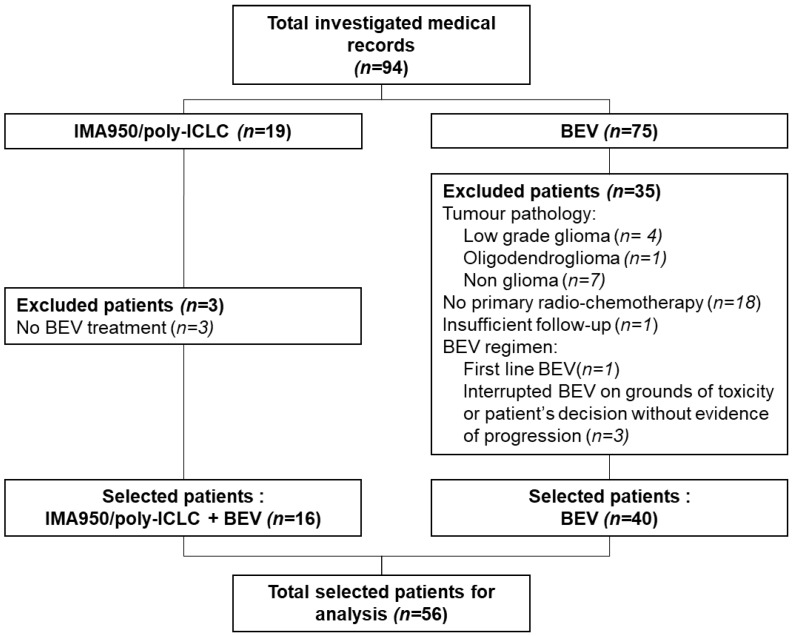
Patient CONSORT diagram. Poly-ICLC, polyinosinic-polycytidylic acid stabilized with polylysine and carboxymethylcellulose; BEV, bevacizumab.

**Figure 2 cancers-11-00464-f002:**
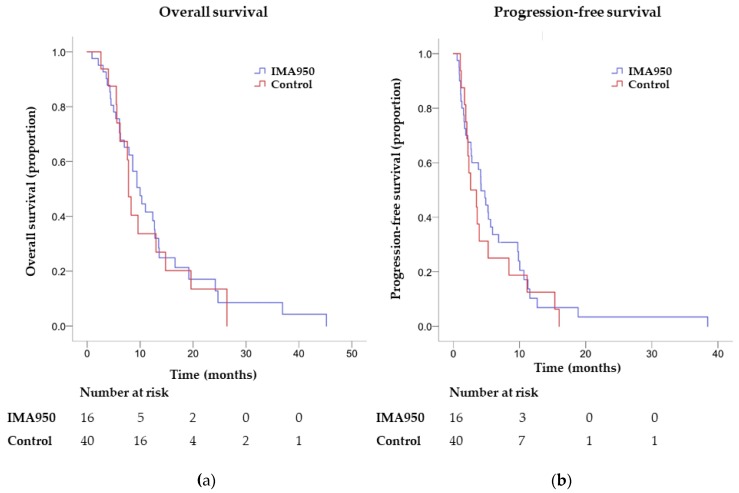
Survival analysis from BEV initiation in control versus IMA950 patients. (**a**) Overall survival (*p* = 0.69); (**b**): Progression-free survival (*p* = 0.50).

**Figure 3 cancers-11-00464-f003:**
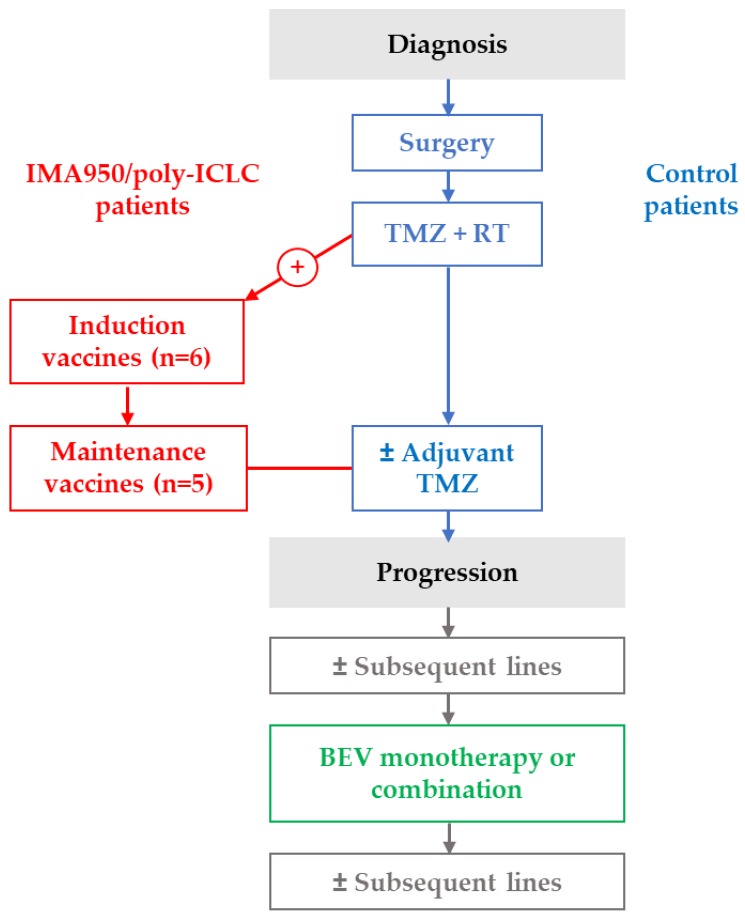
Overview of treatment timelines. RT, radiotherapy; TMZ, temozolomide.

**Table 1 cancers-11-00464-t001:** Baseline patient characteristics.

Variable	IMA950 (*n* = 16)	Control (*n* = 40)	*p*-Value
**Patient variables**			
Median age at BEV initiation (range)	58 (24, 74)	59.5 (33, 76)	0.42
Gender, *n* (%)			0.47
Male	12 (75.0)	26 (65.0)	
Female	4 (25.0)	14 (35.0)	
**Tumour variables**			
Initial diagnosis, *n* (%)			0.40
Primary GBM	13 (81.3)	34 (85.0)	
Primary GBM + PNC	1 (6.3)	0	
Granular cell astrocytoma	0	1 (2.5)	
AA III	2 (12.5)	5 (12.5)	
MGMT gene promoter status, *n* (%)			0.005
Unmethylated	14 (87.5)	15 (37.5)	0.001
Methylated	2 (12.5)	7 (17.5)	0.65
Partially methylated	0	1 (2.5)	0.52
Unknown	0	17 (42.5)	0.002
IDH gene status, *n* (%)			0.16
Wild type	15 (94.0)	29 (72.5)	
Mutated	1 (6.0)	4 (10.0)	
Unknown	0	7 (17.5)	
**Management**			
Extent of resection, *n* (%)			0.20
Biopsy only	2 (12.5)	5 (12.5)	
Partial resection	0	7 (17.5)	
Subtotal/total resection	14 (87.5)	28 70.0)	
Adjuvant TMZ			
# recipients, *n* (%)	16 (100)	36 (90.0)	0.19
# cycles, median (CI)	6.0 (1-)	4 (0–12)	0.35
BEV			
# BEV injections, median (CI)	10.5 (3-)	10.5 (1–73)	0.77
BEV regimen at initiation, *n* (%)			0.37
Monotherapy	12 (75.0)	25 (62.5)	
Combination therapy	4 (25.0)	15 (37.5)	
Line of treatment, *n* (%)			0.17
Second line	14 (87.5)	29 (70)	
> second line	2 (12.5)	12 (30)	
**RPA groups**, N (%)			
III	4 (25.0)	5 (12.5)	0.25
IV	7 (43.7)	17 (42.5)	0.73
V	5 (31.3)	18 (45.0)	0.35

BEV, Bevacizumab; RPA, Recursive partition analysis; GBM, Glioblastoma; GBM + PNC, Glioblastoma with primitive neural cell component; AA, Anaplastic astrocytoma; MGMT, O6-methylguanine-DNA-methyltransferase; IDH, Isocitrate dehydrogenase.

**Table 2 cancers-11-00464-t002:** Survival analysis.

Outcome	IMA950	Control	*p*-Value
**From BEV initiation**			
OS, median (CI), months	7.8 (6.9–8.7)	10.0 (7.4–12.6)	0.69
PFS, median (CI), months	2.6 (0.4–4.9)	4.2 (2.8–5.5)	0.50
PFS-6, *%*	25	30	0.71
PFS-9, *%*	18.8	22.5	0.76
PFS-12, *%*	12.5	7.5	0.55
**From initial diagnosis**			
OS, median (CI), months	19.0 (18.0–20.0)	25.0 (21.6–28.4)	0.53

OS, overall survival; PFS, progression-free survival until end of study; PFS-6, 9, 12, progression free survival rate at 6, 9 and 12 months.

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
