# Peer review of "Exploratory Study of the Effect of IMA950/Poly-ICLC Vaccination on Response to Bevacizumab in Relapsing High-Grade Glioma Patients"

_cancers, 2019, doi:10.3390/cancers11040464_

Round 1

Reviewer 1 Report

In this manuscript entitled “Exploratory study of the effect of IMA950/poly-ICLC vaccination on response to bevacizumab in relapsing high-grade glioma patients”, Boydell et al. investigated whether there is a synergic effect of the treatment of bevacizumab (BEV) and peptide vaccine IMA950/poly-ICLC in high-grade glioma patients. It is no doubt that effective combinatorial treatment is an unmet need for the treatment of GBM patients, such as BEV in combination with immunotherapy. Although the authors designed this study to test synergistic effect of IMA950/poly-ICLC vaccination followed by BEV treatment, there are several major issues that may impair the conclusions. First, the sample size is too small to evaluate this synergistic effect, particularly in IMA950 cohort. Second, the authors should provide evidence for patients’ response to IMA950/poly-ICLC vaccination; third, the authors may provide other factors, e.g. patient life of quality, besides OS because BEV has been shown no benefit for overall survival in high-grade glioma patients. Overall, this study should be significantly improved before being considered for publication.

Author Response

1.      First, the sample size is too small to evaluate this synergistic effect, particularly in IMA950 cohort.

We agree with the reviewer that the number of patients included in the IMA950 trial and receiving bevacizumab is small (16 patients of the initial 19 patients). We entirely agree that this is a major limitation of our study, and had highlighted this in the discussion. This was a post-hoc analysis and therefore not designed with enough statistical power, but rather relied on the existence of a vaccine-treated cohort, whose patients largely went on to receive bevacizumab upon progression. It was indeed this observation, in line with the standard of care in Switzerland, that incited the design of this post-hoc analysis in the direction of evaluating potential synergy, as this represented a unique opportunity (highlighted with the addition of a sub-section entitled “Study design” in the Methods section, lines 234-239). The reason for the small number of patients included in the IMA950 vaccine study stems from the type of study performed. Vaccine trials rely on a thorough immunomonitoring in order to gain knowledge about vaccination efficacy, as the latter can usually not be deduced from clinical outcome. Performing thorough immunomonitoring in a validated way usually implies that small numbers of patients are included (see for example Okada et al. JCO 2011, Pollack et al. JCO 2014, Hilf et al. Nature 2019). We are fully aware that the results of our study should be confirmed with a larger number of patients. We have further highlighted this point in the discussion on lines 228-229 of the manuscript.

2.      Second, the authors should provide evidence for patients’ response to IMA950/poly-ICLC vaccination.

The paper reporting results of the IMA950 vaccination trial being already published (Migliorini et al., Neuro-Oncol 2019), we did not insert results of the immune response to the vaccine in the current manuscript, to prevent duplicate publication. We have now inserted a sentence presenting these results in a summarized form in the discussion (lines 216-217). In addition, we did not find any correlation between immune responses to the IMA950 vaccine and OS or PFS after bevacizumab initiation. This has been added to the discussion section on lines 217-219 of the manuscript.

3.      Third, the authors may provide other factors, e.g. patient life of quality, besides OS because BEV has been shown no benefit for overall survival in high-grade glioma patients. Overall, this study should be significantly improved before being considered for publication.

We thank the reviewer for this suggestion. This is an important point. We are aware that bevacizumab is known to improve quality of life; however, we did not prospectively assess quality of life in the IMA950 trial. Retrospectively assessing quality of life in a credible way in the IMA950 and control patients is not feasible. In addition, bevacizumab has been reported to prolong of PFS, which we tested in the current study, without observing PFS benefit.

Reviewer 2 Report

The manuscript by Boydell and Coll. reports the effect of a double treatment for glioblastoma relapse in a series of patients enrolled at a single center, the Hospital of Geneva University (CH). The manuscript is well written, properly documented and discussed. The study design appears appropriate and the experimentation is accurately executed. Unfortunately for the patients, the double treatment does not confer an advantage on survival compared to bevacizumab alone, yet this negative result is worthy to be known and should be published. 

I have only curiosity, if possible to be mentioned in the text: what was the patients’ response to vaccination? Was this controlled?

Lastly, a very minor comment on the text: at line 259, the words ‘depending on’ appear repeated twice.

Author Response

1.      I have only curiosity, if possible to be mentioned in the text: what was the patients’ response to vaccination? Was this controlled?

We have not reported patient’s response to vaccination in the current manuscript as a paper reporting these results has been recently published (Migliorini et al., Neuro-Oncol 2019). We have now inserted a sentence presenting these results in a summarized form in the discussion (lines 216-217). We additionally describe that we do not find any correlation between immune responses to the IMA950 vaccine and OS or PFS after bevacizumab initiation (lines 217-219 of the manuscript).

2.      Lastly, a very minor comment on the text: at line 259, the words ‘depending on’ appear repeated twice.

We thank the reviewer for pointing out this mistake and have corrected it.

Reviewer 3 Report

cancers-456268

Exploratory study of the effect of IMA950/poly-ICLC vaccination on response to bevacizumab in relapsing high-grade glioma patients

Emma Boydell, Eliana Marinari, Denis Migliorini, Pierre-Yves Dietrich, Anna Patrikidou and Valérie Dutoit

During the last years, various vaccines targeting high-grade glioma have been reported. Among them, some clinical trials have tested, and are currently testing, the use of peptide-based cancer vaccine targeting different glioma-associated antigens alone or combined with immunomodulators. IMA950 was designed as a multipeptide glioblastoma-specific vaccine, with the purpose of activating the immune system targeting tumor cells. Additionally, Polylysine and carboxymethylcellulose (poly-ICLC), a double-stranded RNA (dsRNA), was used at high doses as modulator of the immune system. Nevertheless, it has been reported that at low dose poly-ICLChas less toxicity and results in a broader host defense stimulation. Additionally, it is well known that vascular endothelial growth factor (VEGF), a highly upregulated growth factor in many in high-grade gliomas, can contribute to tumor-associated immunosuppression.

Comments:

Here, the authors describe the study of the effect of IMA950/poly-ICLC vaccination on response to bevacizumab in relapsing high-grade glioma patients. Previously, they have reported the results obtained after the phase I/II trial testing safety and immunogenicity of the multipeptide IMA950/poly-ICLC vaccine in newly diagnosed adult malignant astrocytoma patients, showing that the combination of IMA950 and poly-ICLC was safe. Although the sample size is small, the study is of interest for the field. However, the authors should provide further details regarding the experimental design of the study and methodology (e.g. doses used).

Author Response

However, the authors should provide further details regarding the experimental design of the study and methodology (e.g. doses used).

We thank the reviewer for this comment and have added further information on the IMA950 vaccine composition and doses used on lines 249-253 of the manuscript. The doses of bevacizumab were already described in the Methods section; we have now added details on protocol modifications.

Round 2

Reviewer 1 Report

N/A